# Size-Dependent Effect of Silver Nanoparticles on the Tumor Necrosis Factor α-Induced DNA Damage Response

**DOI:** 10.3390/ijms20051038

**Published:** 2019-02-27

**Authors:** Alaa Fehaid, Akiyoshi Taniguchi

**Affiliations:** 1Cellular Functional Nanobiomaterials Group, Research Center for Functional Materials, National Institute for Materials Science, 1-1 Namiki, Tsukuba, Ibaraki 305-0044, Japan; ahmed.alaa@nims.go.jp; 2Graduate School of Advanced Science and Engineering, Waseda University, 3-4-1 Okubo, Shinjuku-ku, Tokyo 169-8555, Japan; 3Forensic Medicine and Toxicology Department, Faculty of Veterinary Medicine, Mansoura University, Dakahlia 35516, Egypt

**Keywords:** silver nanoparticles, tumor necrosis factor, DNA damage, TNFR1

## Abstract

Silver nanoparticles (AgNPs) are widely used in many consumer products due to their anti-inflammatory properties. Therefore, the effect of exposure to AgNPs should be investigated in diseased states in addition to healthy ones. Tumor necrosis factor-α (TNFα) is a major cytokine that is highly expressed in many diseased conditions, such as inflammatory diseases, sepsis, and cancer. We investigated the effects of two different sizes of AgNPs on the TNFα-induced DNA damage response. Cells were exposed to 10 and 200 nm AgNPs separately and the results showed that the 200 nm AgNPs had a lower cytotoxic effect with a higher percent of cellular uptake compared to the 10 nm AgNPs. Moreover, analysis of reactive oxygen species (ROS) generation and DNA damage indicated that TNFα-induced ROS-mediated DNA damage was reduced by 200 nm AgNPs, but not by 10 nm AgNPs. Tumor necrosis factor receptor 1 (TNFR1) was localized on the cell surface after TNFα exposure with or without 10 nm AgNPs. In contrast, the expression of TNFR1 on the cell surface was reduced by the 200 nm AgNPs. These results suggested that exposure of cells to 200 nm AgNPs reduces the TNFα-induced DNA damage response via reducing the surface expression of TNFR1, thus reducing the signal transduction of TNFα.

## 1. Introduction

Nanotechnology is an advanced field that studies very small materials ranging from 0.1 to 100 nm [1]. Silver nanoparticles (AgNPs) are a high-demand nanomaterial for consumer products [2]. Because of their potent antimicrobial activity, AgNPs are incorporated into many products such as textiles, paints, biosensors, electronics, and medical products including deodorant sprays, catheter coatings, wound dressings, and surgical instruments [3,4,5,6]. Most of the medical applications create concerns over human exposure, due to the properties of AgNPs which allow them to cross the blood brain barrier easily [7].

The characteristics of AgNPs, including morphology, size, size distribution, surface area, surface charge, stability, and agglomeration, have a significant impact on their interaction with biological systems [8,9,10]. All of these physicochemical characteristics affect nanoparticle–cellular interactions, including cellular uptake, cellular distribution, and various cellular responses such as inflammation, proliferation, DNA damage, and cell death [11,12,13]. Therefore, to address safety and improve quality, each characteristic of AgNPs should be clearly determined and separately assessed for its effects on different cellular responses. In this study, we focused on the effect of AgNP size on the cellular response.

Several research groups have investigated the effects of AgNPs with sizes ranging from 5 to 100 nm on different cell lines; the cytotoxic effect of AgNPs on human cell lines (A549, SGC-7901, HepG2, and MCF-7) is size-dependent, with 5 nm being more toxic than 20 or 50 nm and inducing elevated reactive oxygen species (ROS) levels and S phase cell cycle arrest [14]. In RAW 264.7 macrophages and L929 fibroblasts, 20 nm AgNPs are more potent in decreasing metabolic activity compared to the larger 80 and 113 nm nanoparticles, acting by inhibiting stem cell differentiation and promoting DNA damage [15]. Because of the importance of nanoparticle size and its impact on cellular uptake and response, in this study we hypothesized that larger AgNPs with sizes above 100 nm might induce different cellular responses than those of less than 100 nm because of different cellular uptake ratios and mechanisms. Therefore, we investigated the size-dependent effect of AgNPs on a lung epithelial cell line in vitro to elucidate the molecular mechanisms underlying the pulmonary cellular response.

To increase applications for AgNPs, we should consider their effects on diseased subjects as well as healthy ones. Inflammatory diseases, asthma, infections, and cancer are common diseases for which the effects of exposure to AgNPs should be investigated. Tumor necrosis factor-α (TNFα), a pro-inflammatory cytokine and a regulator of immunological reactions in many physiological and pathological conditions [16], is a common molecule that is enhanced in most diseased conditions. TNFα is involved in many signal transduction pathways, such as NF-KB activation, MAPK activation, and cell death signaling, resulting in different cellular responses such as inflammation, DNA damage, proliferation, differentiation, and cell death [17,18,19]. TNFα cellular responses are mainly mediated by one of the two tumor necrosis factor (TNF) receptors (TNFR1 and TNFR2), which elicit different intracellular signals and are without any significant domain homology [20,21]. DNA damage is a very important response because it regulates the cell fate toward death, proliferation, or carcinogenesis; TNFα-induced DNA damage is mostly oxidative and mediated by ROS generation in many cell types [22]. In this study, we hypothesized that AgNPs affect DNA damage along with their known anti-apoptotic and anti-inflammatory effects, so we focused on the TNFα-induced DNA damage response.

We investigated the size-dependent effect of AgNPs, and our results revealed that the expression of TNFR1 on the cell surface was reduced by 200 nm AgNPs but not by 10 nm AgNPs, suggesting a reduction in TNFα-induced DNA damage by 200 nm AgNPs.

## 2. Results

### 2.1. Effect of AgNPs on Cell Viability

The size of AgNPs is one of their most important characteristics and influences their uptake by cells and the cellular response. Our aim was to clarify the size-dependent cytotoxic effect of AgNPs. Many studies have investigated the effect of AgNPs in particle sizes ranging from 10 to 100 nm; however, nanoparticles larger than 100 nm might have different effects because they can induce different mechanisms of cellular uptake or have a different uptake ratio. We therefore conducted a cell viability assay to determine the differences between 10 nm and 200 nm AgNPs on the viability of NCI-H292 cells. As shown in Figure 1, the percentage of viable cells decreased in a dose-dependent manner in cells exposed to 10 nm and 200 nm AgNPs (increasing the concentration of AgNPs reduced the percentage of viable cells). Cells exposed to 200 nm AgNPs showed lower cytotoxic effects compared to the 10 nm AgNP-exposed cells; the percentages of viable cells after 24 h exposure to 1, 2.5, 5, 10, 25, 50, 75, and 100 µg/mL of 200 nm and 10 nm AgNPs were 110.1%, 109.8%, 109.3%, 107.2%, 98.2%, 87.4%, 74.5%, and 73.1%; and 98.2%, 99.7%, 94.2%, 86,1%, 59.9%, 38.8%, 29.4%, and 26.2%, respectively. These results demonstrated that the 200 nm AgNPs had a lower cytotoxic effect than the 10 nm AgNPs, showing the impact of nanoparticle size on cytotoxicity.

### 2.2. Cellular Uptake of AgNPs

Cellular uptake of nanoparticles plays an important role in cellular responses including proliferation, inflammation, DNA damage, and cell death. We therefore estimated the cellular uptake of 10 nm and 200 nm AgNPs, and the results are shown in Figure 2. The percentage of cells incorporated with 200 nm AgNPs was higher than the percentage of cells incorporated with 10 nm AgNPs, resulting in an increase in cell density as expressed by side scatter (SSC) as shown in the right panel of Figure 2A. After 24 h of exposure, uptake of 200 nm AgNPs occurred in 30.5% of cells, while uptake of 10 nm AgNPs occurred in only 11.5% of cells, as shown in Figure 2B. These results revealed that larger AgNP size (200 nm) induced higher cellular uptake than a smaller size (10 nm).

### 2.3. Interference of AgNPs with TNFα-Induced ROS Generation

In many disease states such as inflammatory disease, infections, and cancer, TNFα acts as a major cytokine. TNFα has been reported to be involved in ROS generation resulting in DNA damage and cell death [23]. Therefore, we conducted a DCF assay to understand how different sizes of AgNPs affect TNFα-induced ROS generation. As shown in Figure 3, cells exposed to TNFα (20 ng/mL) only, 10 nm AgNPs (100 µg/mL) only, or both showed highly significant increases in ROS generation compared to the negative control group. Moreover, cells exposed to TNFα (20 ng/mL) + 200 nm AgNPs (100 µg/mL) showed a significant decrease in ROS generation compared to the TNFα-exposed group. These data suggested that the 200 nm AgNPs, but not the 10 nm AgNPs, reduced TNFα-induced ROS generation. Also, only 10 nm AgNPs induced ROS generation on their own.

### 2.4. Effect of AgNPs on TNFα-Induced DNA Damage

ROS-mediated DNA damage is well known to be induced by TNFα. Since TNFα-induced ROS generation was affected by AgNPs, DNA damage was also evaluated as an important cellular response that is influenced by ROS generation and regulates cell fate.

For this purpose, we transfected a B-cell Translocation Gene 2 (*BTG2*) promoter-reporter plasmid to detect the DNA damage response. As shown in Figure 4, the fold inductions of the *BTG2* response in cells exposed to TNFα (20 ng/mL) only, 10 nm AgNPs (100 µg/mL) only, or both, were significantly increased compared to the control group. However, cells exposed to TNFα (20 ng/mL) + 200 nm AgNPs (100 µg/mL) showed a significant decrease in the fold induction of the response compared to the TNFα-exposed group. These results suggest that 200 nm AgNPs might regulate TNFα-induced DNA damage.

DNA damage is regulated by the DNA damage response (DDR) signaling pathway, which uses signal sensors, transducers, and effectors; the ataxia-telangiectasia mutated (ATM) and ATM- and Rad3-Related (ATR) proteins are the most upstream DDR kinases that are activated by the sensors of the DDR pathway [24]. Therefore, we used PCR array to detect the regulation of genes associated with ATM/ATR signaling as a featured pathway for DDR. Genes that were upregulated by more than 1-fold are listed in Table 1. In particular, the expressions of *ATM*, the CHK1 checkpoint homolog of *Schizosaccharomyces pombe* (*CHEK1*), the RAD21 homolog of *S. pombe* (*RAD21*), structural maintenance of chromosomes 1A (*SMC1A*), and tumor protein p53 (*TP53*) genes were increased by ≥ 1.9-fold in cells exposed to TNFα (20 ng/mL). Also, these same genes exhibited ≥ 1.2-fold induction in cells exposed to TNFα (20 ng/mL) + 10 nm AgNPs (100 µg/mL). However, in cells exposed to TNFα (20 ng/mL) + 200 nm AgNPs (100 µg/mL), the expression of these genes showed downregulation of ≤ 0.8-fold, indicating a reduction in TNFα-induced DNA damage by 200 nm AgNPs. To confirm the induction of the above five genes, we conducted real-time PCR analysis. As shown in Figure 5, none of the genes in cells exposed to TNFα + 10 nm AgNPs showed any significant difference in expression compared to the TNFα-exposed group, except for *SMC1A*, which showed a significant decrease from 2.5- to 1.8-fold induction. In contrast, for cells exposed to TNFα (20 ng/mL) + 200 nm AgNPs (100 µg/mL), all five genes showed significant downregulated expression compared to the TNFα-exposed group, especially *TP53*, *RAD21*, and *CHEK1*, which were downregulated from 2.5 to 0.9, 1.6 to 0.4, and 2.3 to 0.5, respectively. The mRNA expressions of these genes involved in the DDR signaling demonstrated that most of the TNFα-induced upregulated genes were downregulated by 200 nm but not 10 nm AgNPs, suggesting that 200 nm AgNPs reduced the TNFα-induced DNA damage.

### 2.5. Localization of Tumor Necrosis Factor-α Receptor 1 (TNFR1)

TNFα has two receptors, TNFR1 and TNFR2. TNFR1 is the major receptor, exists in most cell types, and mediates the NF-KB activation signaling pathway, which is involved in ROS generation and DNA damage [25]. Our above-mentioned results indicated that the 200 nm AgNPs, but not the 10 nm AgNPs, affected TNFα-induced ROS production and DNA damage. Also, the cellular uptake of the 200 nm was higher than that of the 10 nm AgNPs. We hypothesized that TNFR1 might have a role in this effect. We examined the localization of TNFR1 by immunofluorescence staining using confocal microscopy as shown in Figure 6. The results revealed that TNFR1 is slightly aggregated and distributed on the membranes of TNFα-exposed cells as shown in Figure 6a. Also, it was distributed over the entire membrane of cells exposed to TNFα (20 ng/mL) + 10 nm AgNPs (100 µg/mL) as shown in Figure 6b. In contrast, Figure 6c shows that TNFR1 was localized inside the cells with very few localized on the membranes of cells exposed to TNFα (20 ng/mL) + 200 nm AgNPs (100 µg/mL). These data suggest that 200 nm AgNPs reduced the expression level of TNFR1 on the cell membrane, and this reduction in surface expression of TNFR1 reduced the signal transduction of TNFα, resulting in a reduction in TNFα-induced DNA damage.

## 3. Discussion

AgNPs are considered to be a double-edged sword that can induce opposing effects. AgNPs have a well-known potential anti-inflammatory effect [26,27], but they can also induce inflammatory responses [28,29,30]. Moreover, our previous research found an anti-apoptotic effect of AgNPs [31], while some other reports have found that AgNPs can induce apoptosis [32,33]. The size of AgNPs is one of the most important characteristics that modulates their opposing effects. Therefore, size should be clearly determined, and each effect specified for each size. Generally, after the internalization of AgNPs into cells, many different cellular responses are seen such as proliferation, inflammation, DNA damage, and cell death. The determination of specific cellular responses to specific sizes would provide better details about the molecular mechanisms of the induced responses.

Here, we investigated the size-dependent effects of polyvinylpyrrolidone (PVP)-coated AgNPs. We used 10 and 200 nm particles, hypothesizing that they would have different behaviors when interacting with lung epithelial cells. Interestingly, our results showed that the 200 nm particles were less cytotoxic (Figure 1), despite the significant increase in their cellular uptake (Figure 2) compared to the 10 nm AgNPs. These results suggest that thorough uptake of the 200 nm particles by cells might occur via endocytosis of their spheres, and while being held in endosomes they are not easily ionized, which results in their low cytotoxic effect. In contrast, uptake of the 10 nm AgNPs occurred easily through the cell membrane to the cytoplasm. However, the cytoplasmic environment would enhance the ionization of AgNPs, allowing the Ag ions to induce a strong cytotoxic effect. By the same mechanism, the results shown in Figure 3 indicated that ROS generation in cells exposed to 10 nm AgNPs was significantly increased compared to control cells because of this ionization. Dissolution of AgNPs and ion release are always related to their cytotoxicity; it has been found that the smaller nanoparticles are more toxic because of their larger surface area which induces faster dissolution and ion release [34,35]. On the other hand, the PVP coating of AgNPs could increase the stability of the nanoparticles (NPs) and reduce the amount of released Ag ions in the culture medium [36]. Therefore, the difference in the produced cytotoxic effect of 10 nm and 200 nm AgNPs could be due to a combination of both ion release from the nanoparticles and different ways of cellular uptake and uptake ratios.

TNFα is highly expressed and is involved in many acute and chronic inflammatory diseases and cancer; it also induces many different signal transduction pathways that regulate cellular responses [37,38]. Since our goal was to investigate the effects of exposure to different sizes of AgNPs under diseased states, we used TNFα as a DNA damage-inducing agent. The relationship between AgNPs of different sizes and the TNFα-induced DNA damage response was analyzed. The results of DNA damage analysis by *BTG2* response (Figure 4), gene expression by PCR array (Table 1), and RT-PCR (Figure 5) were all consistent with the ROS generation after exposure of the cells to 10 and 200 nm AgNPs. All results confirmed that the 200 nm AgNPs reduced TNFα-induced DNA damage. In contrast, 10 nm AgNPs could induce DNA damage by their own action without affecting that induced by TNFα. These results suggest that the 200 nm AgNPs can reduce DNA damage in diseased conditions that occurs via TNFα.

In order to understand the molecular mechanism of the change in TNFα-induced DNA damage response by the differently sized AgNPs, TNFR1 localization was determined by confocal microscopy. TNFR1 is a receptor of TNFα, and when they bind together TNFα signal transduction is induced. Therefore, TNFR1 might play a role in the different effects of the 10 and 200 nm AgNPs. As shown in Figure 6, in cells exposed to TNFα only, TNFR1 was distributed on the cell membrane surface with few aggregations. Also, in cells exposed to TNFα and 10 nm AgNPs together, TNFR1 was distributed homogenously on the cell membrane. In contrast, TNFR1 was localized mainly inside cells with very few receptors scattered on the membrane surface during exposure to both TNFα and 200 nm AgNPs. These results prompted us to propose the molecular mechanism shown in Figure 7. In cells exposed to TNFα only, TNFα specifically binds to TNFR1 by receptor/ligand binding, and they move together into cells to release TNFα and free the receptors to return to the cell membrane to bind more TNFα. This normal binding cycle induces TNFα signal transduction, leading to the ROS-mediated DNA damage response. However, in cells exposed to both TNFα and 200 nm AgNPs, the nanoparticles might attach to TNFR1/TNFα to form a TNFR1/TNFα/NP complex, which is then endocytosed inside of cells by receptor-mediated endocytosis. TNFα would then be easily released from the receptor and induce signal transduction, while TNFR1 might remain in complex with the 200 nm AgNPs. This complex might change the receptor properties such as shape, molecular weight, and surface characteristics, resulting in a disturbance of the receptor’s normal pathway of returning to the cell membrane, causing less TNFR1 localization on the cell membrane and more inside cells, as shown in Figure 6C. In cells exposed to both TNFα and 10 nm AgNPs, TNFα enters cells by normal TNFR1/TNFα binding. Then, TNFα is released from the receptors and they return to the surface of the cell membrane. Meanwhile, the 10 nm AgNPs uptake by cells occurs through the membrane without any receptor-dependent uptake, so they do not affect the localization of TNFR1 on the membrane surface, as shown in Figure 6B. This mechanism would explain the role of TNFR1 in the size-dependent effect of AgNPs on TNFα-induced DNA damage. It indicates that the 200 nm AgNPs hinder recycling of the TNFR1 to the cell membrane, resulting in a decrease in TNFα signal transduction followed by a decrease in the DNA damage response. In contrast, the 10 nm AgNPs have no effect on localization of TNFR1 to the cell membrane. Therefore, TNFα signal transduction and DNA damage are not affected by 10 nm AgNPs. This indicates an independent mode of action for 10 nm AgNPs.

In this study, we investigated the cellular response of the lung epithelial cell line after exposure to TNFα and AgNPs. However, AgNPs not only affect the epithelial cells but also induce changes in the cellular responses of the immune cells specially the macrophages [39,40], therefore, we suggest further testing of the effect of AgNPs on the cellular responses of TNFα in macrophage cell lines.

In addition, the properties of TNFα imply that TNFα blockers are useful as a therapy for many different diseases like Alzheimer’s disease [41] or as an adjuvant for cancer treatment [42]. There are currently successful applications in the treatment of chronic inflammatory diseases such as rheumatoid arthritis [43,44] using TNFα blockers. Our findings suggest that 200 nm AgNPs could serve as a promising TNFα blocker. Further in vivo testing is needed to discover their therapeutic potential as a new strategy to block TNFα using a laboratory animal model of inflammatory diseases to support our in vitro findings.

## 4. Materials and Methods

### 4.1. Cell Culture

Human pulmonary epithelial cell line NCI-H292 (ATCC CRL-1848TM) cells were cultured in an incubator with a humidified atmosphere containing 5% CO2 at 37°C. RPMI-1640 medium (L-glutamine with phenol red, Nacalai Tesque, Japan) supplemented with 10% (*v*/*v*) heat-inactivated fetal bovine serum (HFBS, Biowest, USA), 100 μg/mL penicillin, and 10 μg/mL streptomycin (Nacalai Tesque) was used to culture the cells.

### 4.2. Silver Nanoparticles (AgNPs)

Polyvinylpyrrolidone (PVP)-coated AgNPs with two different sizes of 10 nm and 200 nm (Cat. Nos. 795925 and 796026, respectively; Sigma-Aldrich, USA) were used in this study. Electronic light scattering (zeta potential and particle size analyzer ELSZ-2000, Otsuka Electronics, Japan) was used to analyze the particle sizes and zeta potentials. The average hydrodynamic diameter of 10 nm AgNPs in deionized water was 12.0 ± 1.8 nm, the polydispersity index was 0.191, and the zeta potential was −21.45 mV. For the 200 nm AgNPs, the average hydrodynamic diameter in deionized water was 221.9 ± 50.9 nm, the polydispersity index was 0.026, and the zeta potential was −27.59 mV. AgNP suspensions were concentrated and sterilized by autoclaving (121 °C for 20 min), then the working solutions were prepared by resuspension in RPMI 1640 medium for cell exposure.

### 4.3. Tumor Necrosis Factor-α (TNFα)

Recombinant human TNFα (Peprotech, USA) was reconstituted in water to 100 µg/mL. The working dilutions were prepared using sterilized culture medium (RPMI-1640) containing a carrier protein.

### 4.4. Cell Viability Assay

The viability of NCI-H292 cells was measured using a CellTiter-Glo^®^ luminescent cell viability assay (Promega, Madison, WI, USA) according to the manufacturer’s protocol. Cells were seeded at a density of 1 × 104 cells/well in an opaque 96-well plate and incubated at 37 °C and 5% CO_2_ overnight, then the cells were checked for 80–90% confluency. Cells were exposed to 10 nm and 200 nm AgNPs (final concentrations of 0, 5, 10, 25, 50, 75, and 100 µg/mL) separately for 24 h. CellTiter-Glo^®^ reagent was added to each well, and the percent of viable cells was calculated based on quantification of adenosine triphosphate (ATP) using a luminometer (TECAN, Japan).

### 4.5. Cellular Uptake Assay

To check the percentage of cells that incorporated AgNPs, NCI-H292 cells were seeded at a concentration of 8 × 104 cells/well in a 24-well plate (Costar, Washington, DC, USA). After overnight incubation, the cells were exposed to a final concentration of 100 µg/mL of both 10 nm and 200 nm AgNPs. After 24 h of exposure, the cells were washed twice with phosphate buffered saline (PBS) and collected by trypsinization using trypsin EDTA (Wako, Japan) and centrifugation. Then, the cells were resuspended in 1 mL PBS supplemented with 6% HFBS and kept on ice until analysis.

The percentage of cells taking up AgNPs was analyzed using a flow cytometer (FACS, SP6800 spectral analyzer, Sony Biotechnology, Japan). Forward scatter (FSC) is the laser light scattered at narrow angles to the axis of the laser beam and is proportional to the cell size. Side scatter (SSC) is the laser light scattered at a 90° angle to the axis of the laser and is proportional to the intracellular density, which is increased by the uptake of nanoparticles. The mean SSC for each group of cells was calculated depending on the peak intensities of treated cells compared to the control cells using the software supplied with the instrument.

### 4.6. DCF Assay for Oxidative Stress Determination

To quantify intracellular ROS, a ROS assay kit (OxiSelectTM, Cell Biolabs, Inc., USA) was used. This assay is based on the cell permeable fluorogenic probe 2′,7′-dichlorodihydrofluorescein diacetate (DCFH-DA), which diffuses into cells and is deacetylated by intracellular esterases to the nonfluorescent 2′,7′-dichlorodihydrofluorescein (DCFH). In the presence of ROS, DCFH is rapidly oxidized to the highly fluorescent 2′,7′-dichlorodihydrofluorescein (DCF). The fluorescence intensity is proportional to the intracellular ROS levels.

According to the manufacturer’s protocol, cells were cultured at a density of 1 × 104 cells/well in a black 96-well plate and incubated overnight. Subsequently, media were removed, and cells were washed twice gently with DPBS (14190, GIBCO, Invitrogen, Carlsbad, CA, USA) and incubated with DCFH-DA/media solution for 30 min in the dark at 37 °C. Then, after removing the solution and washing the cells with DPBS, the DCFH-DA-loaded cells were exposed to TNFα (20 ng/mL) and 10 nm AgNPs (100 µg/mL) or 200 nm AgNPs (100 µg/mL) separately and together for 24 h. Parallel sets of wells containing DCFH-DA-loaded cells without any further exposure were used as a negative control. Another set of DCFH-DA-loaded cells were exposed to hydrogen peroxide (H_2_O_2_) and used as a positive control. The fluorescence of DCF was measured at regular intervals at an excitation/emission wavelength of 480 nm/530 nm using a fluorometric plate reader (Microplate Fluorometer, Twinkle LB 970, BERTHOLD TECHNOLOGIES, BadWildbad, Germany). The amounts of produced DCF were calculated based on a DCF standard curve.

### 4.7. Dual-Luciferase Reporter Assay for *BTG2* Response Assessment

#### 4.7.1. Plasmids Employed

pGL3-Control vector (E1741, Promega) was used as an empty control reporter plasmid. *BTG2* promoter-reporter plasmid (the region from nt −100 to −20 bp of the *BTG2* gene containing a p53 binding site mutation) was used to detect DNA damage. Both reporter plasmids contain SV40 promoters and enhancer sequences that result in strong expression of the luciferase encoding gene (luc+) in different types of mammalian cells. Also, pRL-CMV vector (E2261, Promega), which is a Renilla luciferase-encoding control plasmid, was used as an internal control for variations in the transfection efficiency.

#### 4.7.2. Transfection

pGL3 blank control reporter plasmid or *BTG2* promoter-reporter plasmid and pRL-CMV internal control plasmid were co-transfected into NCI-H292 cells. LipofectamineTM LTX reagent with a PlusTM reagent kit (Invitrogen) was used to perform the transfection according to the manufacturer’s protocol. Cells were cultured at a density of 2 × 105/mL in a 24-well plate and incubated overnight at 37 °C. Opti-MEM medium (Life Technologies, Carlsbad, CA, USA) was used to dilute the Lipofectamine LTX reagent and plasmids, then the Plus reagent was added to the diluted plasmids. Diluted plasmids with Plus reagent were added to the diluted Lipofectamine LTX reagent at 1:1 ratio and incubated for 5 min at room temperature. Finally, the plasmid–lipid complexes were added to the cells and incubated at 37 °C for at least 24 h before exposure to TNFα and AgNPs.

#### 4.7.3. Assessment of Luciferase Activity

After exposure of transfected cells to TNFα (20 ng/mL) and 10 nm AgNPs (100 µg/mL) or 200 nm AgNPs (100 µg/mL) separately and together for 24 h, the luciferase activities were assessed using a Dual-Luciferase Reporter Assay System (E1910, Promega) according to the manufacturer’s protocol. Cells were lysed using 1× passive lysis buffer and gentle shaking for 10 min, then cell lysates were transferred to tubes containing luciferase assay reagent II (LAR II). Firefly luciferase (F) signals were measured, then Stop & Glo reagent was added to the tubes and the Renilla luciferase (R) signals were also measured. The firefly and Renilla luciferase signals were recorded using a luminometer (Lumat LB9507, BERTHOLD TECHNOLOGIES) according to the instrument manual. The changes in luciferase activities were calculated using the following equation:∆ Fold activity = (F/R) sample ÷ (F/R) control.

### 4.8. Gene Expression Analysis

#### 4.8.1. Polymerase Chain Reaction (PCR) Array

To analyze the expression of genes involved in DNA damage, PCR array analysis was conducted as follows. NCI-H292 cells were seeded at a concentration of 4 × 105 cells/60 mm cell culture dish. After overnight incubation, the cells were exposed to TNFα (20 ng/mL) only, or together with 10 nm AgNPs (100 µg/mL) or 200 nm AgNPs (100 µg/mL). After 8 h of exposure, the cells were detached by trypsinization and collected by centrifugation, and then the total cellular RNA was extracted using an RNeasy kit (Qiagen, Germantown, MD, USA) according to the manufacturer’s protocol. An aliquot (1 µg) of the extracted total RNA was reverse transcribed into cDNA using a RT2 First Strand kit (SABiosciences/Qiagen), and the expression of 89 Human DNA Damage Signaling Pathway genes was measured using a RT2 profiler PCR array kit (SABiosciences/Qiagen) according to the manufacturer’s protocol. PCR array analysis was performed using an ABI PRISM 7000 sequence detection system (Applied Biosystems, Singapore, Singapore).

#### 4.8.2. Real-Time (RT) PCR

For mRNA expression analysis, cells were seeded and exposed to TNFα and AgNPs, then total RNA and cDNA were synthetized as mentioned for the PCR array. The PCR primers for human *SMC1A*, *ATM*, *TP53*, *RAD21*, and *CHEK1* were purchased from SABiosciences/Qiagen. The reaction mixture was composed of 12.5 µL RT2 SYBR Green qPCR Master Mix (SABiosciences/Qiagen), 1 µL 10 µM gene-specific RT2 qPCR forward and reverse primers, 2 µL cDNA, and nuclease-free water to a final volume of 25 µL. Glyceraldehyde-3-phosphate dehydrogenase (*GAPDH*) was used as a house-keeping gene to normalize the data. RT-PCR analysis was performed using the same machine used for PCR array, and the thermocycling conditions were 95°C for 10 min, followed by 40 cycles of 95 °C for 15 s and 60 °C for 1 min.

### 4.9. Immunostaining and Confocal Laser Scanning Microscopy

To localize tumor necrosis factor receptor 1 (TNFR1), NCI-H292 cells were seeded in a CELLview cell culture dish (Greiner Bio-one North America Inc., Monroe, NC, USA) at a density of 1.5 × 104 cells/compartment and incubated for 24 h. The cells were exposed to TNFα (20 ng/mL) only, or together with 10 nm AgNPs (100 µg/mL) or 200 nm AgNPs (100 µg/mL). After 24 h of exposure, the cells were washed with 1× PBS fixed with 4% formaldehyde solution in PBS (Wako) at room temperature, permeabilized with 0.1% Triton X-100, and then blocked with 10% normal goat serum in PBS for 1 h. The cells were then incubated overnight at 4°C with rabbit polyclonal anti-TNF receptor 1 antibody (Abcam, Cambridge, UK) followed by incubation with labeled goat anti-rabbit IgG H&L (Alexa Fluor 488) (Abcam) for 1 h at room temperature. Nuclear DNA was stained with DAPI (4′, 6-diamidino-2-phenylindole) (Dojindo, Kumamoto, Japan) for 5 min at room temperature. Microscopic observations and images were acquired using a confocal laser-scanning microscope (LSM510 META, Carl Zeiss Inc., Jena, Germany) with a 63 × 1.4 Plan-Apochromat oil immersion lens.

### 4.10. Statistical Analysis

Statistical analysis was performed using Student’s t-test. Differences and significances between means of different groups were determined using one-way ANOVA with Duncan’s multiple comparison tests. P values less than 0.05 were considered statistically different. Data are presented as means ± standard deviation (SD) with at least three independent replicates (*n* ≥ 3).

## 5. Conclusions

In this study, we found that 200 nm AgNPs, but not 10 nm AgNPs, reduced DNA damage in NCI-H292 cells and proposed a mechanism for this effect. This mechanism works by reducing membrane localization of TNFR1 and thus decreasing TNFα signal transduction, leading to a reduction in TNFα-induced DNA damage. Also, the mechanism explains why 10 nm AgNPs induced ROS-mediated DNA damage by their own action without affecting TNFR1 and TNFα signal transduction.

## Figures and Tables

**Figure 1 ijms-20-01038-f001:**
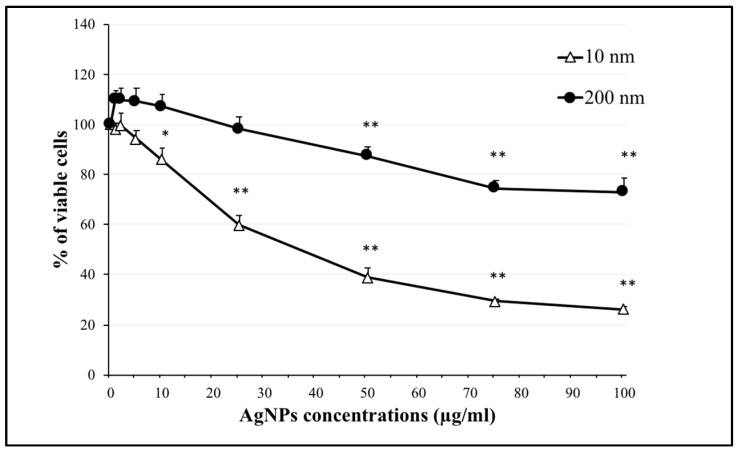
Effect of silver nanoparticles (AgNPs) (10 nm and 200 nm) on the viability of NCI-H292 cells. Viability of cells exposed to 10 nm and 200 nm AgNPs separately at concentrations of 0, 5, 10, 25, 50, 75, and 100 µg/mL. Cells were exposed to AgNPs for 24 h, then cell viability was determined using a CellTiter-Glo^®^ luminescent cell viability assay. The results are shown as means ± SD, n ≥ 3, for each group; *0.01 < *P* < 0.05, ** *P* < 0.01. * Represents significance compared to the control group.

**Figure 2 ijms-20-01038-f002:**
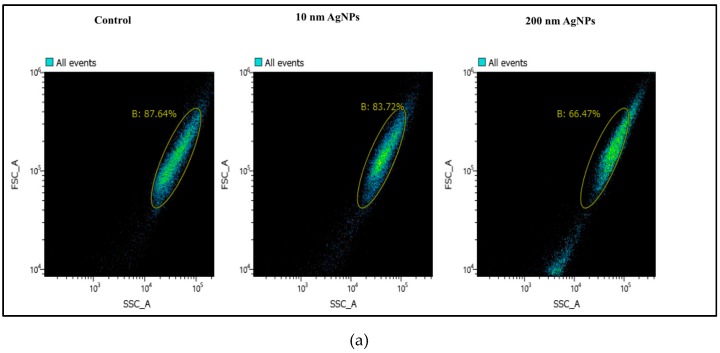
Uptake ratios of 10 nm and 200 nm AgNPs by NCI-H292 cells. Cells were incubated with AgNPs at a concentration of 100 μg/mL for 24 h. Cellular uptake of AgNPs was calculated using FACS based on the side scatter (SSC). (**a**) Gated forward and side scatter plot of most cells depending on the control population; the cells exposed to 200 nm AgNPs showed higher SSC in the right panel. (**b**) Percentage of cells incorporated with 10 nm and 200 nm AgNPs. The results are shown as means ± SD, *n* ≥ 3, for each group; ** *P* < 0.01.

**Figure 3 ijms-20-01038-f003:**
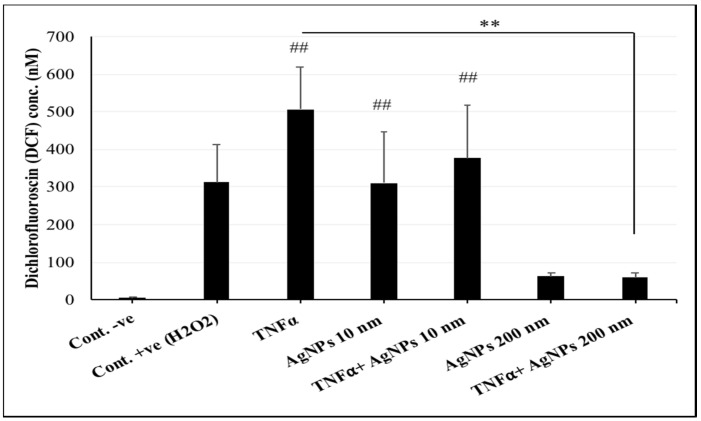
Reactive oxygen species (ROS) production in NCI-H292 cells. Cells were exposed to tumor necrosis factor-α (TNFα) (20 ng/mL) and AgNPs 10 nm (100 µg/mL) or AgNPs 200 nm (100 µg/mL) separately and together for 24 h. ROS production is expressed by the produced DCF amount. The results are shown as means ± SD, *n* ≥ 3, for each group; ## and ** indicate *P* < 0.01. ## represents a significant increase compared to the Control -ve group, ** represents a significant decrease compared to the TNFα-exposed group. Cont. -ve are the non-treated cells and Cont. +ve are the cells exposed to H2O2 to induce ROS generation.

**Figure 4 ijms-20-01038-f004:**
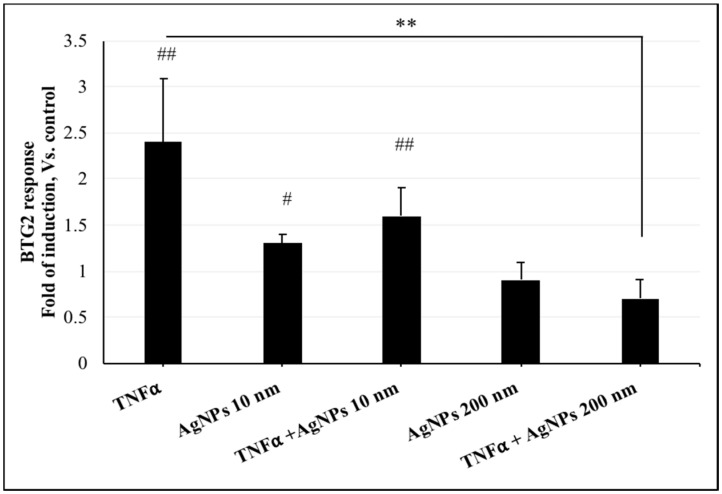
*BTG2* response in NCI-H292 cells. Cells were transfected with *BTG2* promoter-reporter plasmid, and then the transfected cells were exposed to TNFα (20 ng/mL) and AgNPs 10 nm (100 µg/mL) or AgNPs 200 nm (100 µg/mL) separately and together for 24 h. The results are shown as means ± SD, *n* ≥ 3, for each group; # indicates 0.01 < *P* < 0.05, ##, ** indicates *P* < 0.01. # represents a significant increase compared to the control group. * represents a significant decrease compared to the TNFα -exposed group.

**Figure 5 ijms-20-01038-f005:**
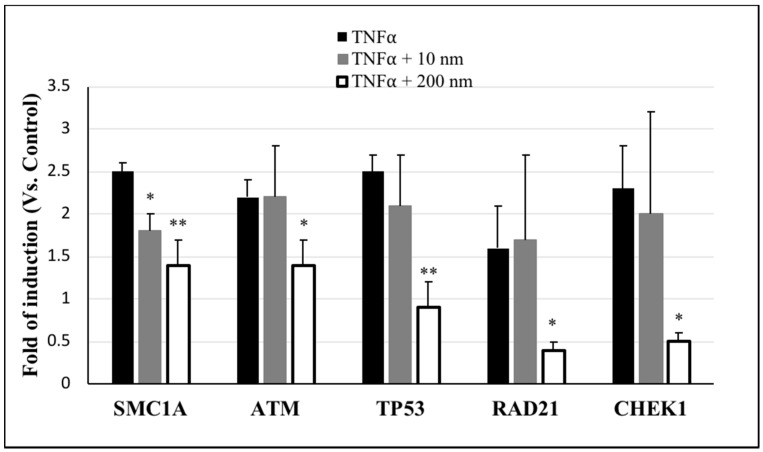
Expression of DNA damage marker mRNAs in NCI-H292 cells. Cells were exposed to TNFα (20 ng/mL) only, TNFα + AgNPs 10 nm (100 µg/mL), or TNFα + AgNPs 200 nm (100 µg/mL) for 8 h. Expressions of mRNAs were measured using RT-PCR. The results are shown as means ± SD, *n* ≥ 3, for each group; *0.01 < *P* < 0.05, ***P* < 0.01, and each represents significant differences compared to the corresponding TNFα-exposed group.

**Figure 6 ijms-20-01038-f006:**
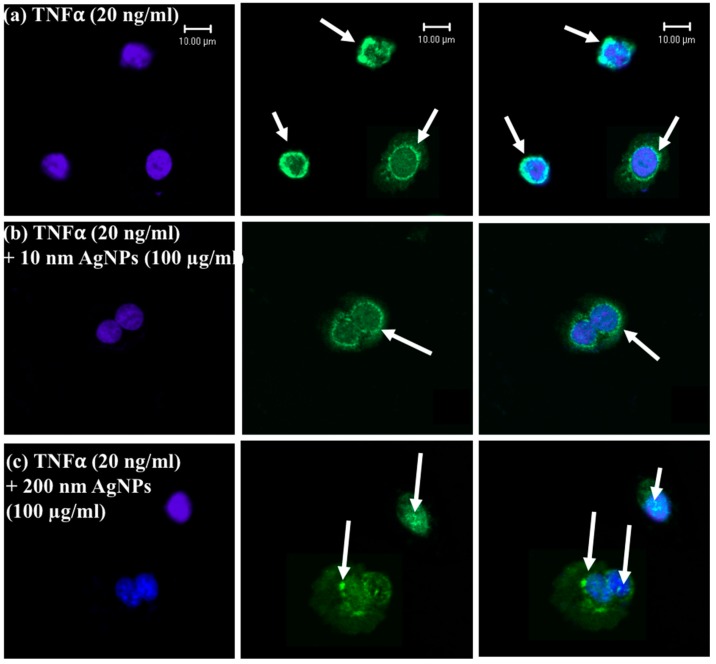
Localization of TNFR1 in NCI-H292 cells using confocal microscopy. Blue shows the nucleus, green shows the receptors (TNFR1), and blue and green together are the merged form. White arrows show TNFR1. (**a**) NCI-H292 cells were exposed to TNFα (20 ng/mL), and TNFR1 was distributed on the cell membrane with some aggregations. (**b**) NCI-H292 cells were exposed to both TNFα (20 ng/mL) + 10 nm AgNPs (100 µg/mL), and TNFR1 localization was scattered over the entire cell membrane. (**c**) NCI-H292 cells were exposed to both TNFα (20 ng/mL) and 200 nm AgNPs (100 µg/mL), and TNFR1 was localized inside cells with very few receptors on the cell membrane. Exposure was 24 h for all experiments. Scale bar is 10 µm for all panels.

**Figure 7 ijms-20-01038-f007:**
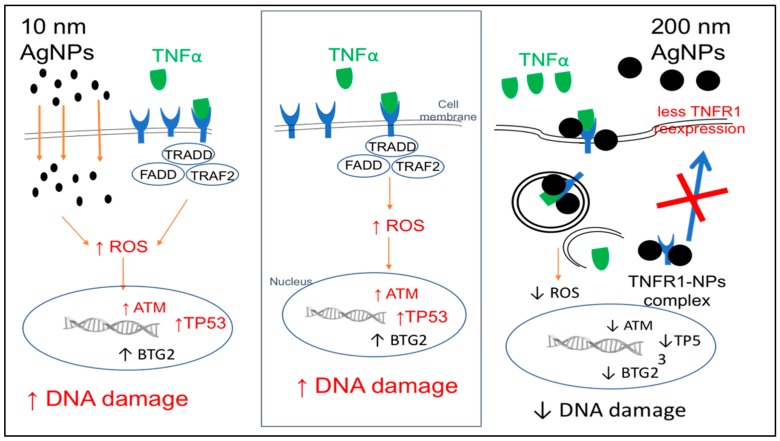
Proposed molecular mechanism explaining why TNFα-induced DNA damage was reduced by 200 nm AgNPs but not by 10 nm AgNPs, and how 200 nm AgNPs decreased membrane localization of TNFR1.

**Table 1 ijms-20-01038-t001:** Induction of mRNA expression of DNA-damage genes in NCI-H292 cells.

Symbol of Genes	Description of the Genes	Fold Regulation Vs. Control		
		TNFα	TNFα + 10 nm AgNPs	TNFα + 200 nm AgNPs
ATM	Ataxia telangiectasia mutated	2.1	1.9	**0.8**
CDK7	Cyclin-dependent kinase 7	3.8	6.9	1
CHEK1	CHK1 checkpoint homolog (*S. pombe*)	2.9	4.4	**0.4**
DDIT3	DNA-damage-inducible transcript 3	2.1	2.7	1.8
RAD21	RAD21 homolog (*S. pombe*)	1.9	2.9	**0.3**
RAD51	RAD51 homolog (*S. cerevisiae*)	1.5	1.6	0.6
SIRT1	Sirtuin 1	1.3	3.5	0.8
SMC1A	Structural maintenance of chromosomes 1A	1.9	1.2	**0.7**
SUMO1	SMT3 suppressor of mif two 3 homolog 1 (*S. cerevisiae*)	2.5	4.1	1.6
TP53	Tumor protein p53	2.6	2.8	**0.6**

Cells were exposed to TNFα (20 ng/mL) only, or TNFα + 10 nm AgNPs (100 µg/mL), or TNFα + 200 nm AgNPs (100 µg/mL) for 8 h. Expressions of mRNAs were measured using a DNA damage RT2 Profiler PCR Array. Fold regulation values more than 1 were considered as positive regulation (upregulation).

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
