# Peer review of "Size-Dependent Effect of Silver Nanoparticles on the Tumor Necrosis Factor α-Induced DNA Damage Response"

_ijms, 2019, doi:10.3390/ijms20051038_

Reviewer 1 Report

The use of Ag will always raise the issue of solubility and the release of Ag ion. This must be discussed. The contribution of ion released is size dependent, it would be nice to estimate this dose. This is the missing link which would explains the observations.

Author Response

Thank you for your comments. Please find the revised manuscript on the web site. Revisions have been made after carefully considering the comments raised by the reviewer.

Reviewer 1

The use of Ag will always raise the issue of solubility and the release of Ag ion. This must be discussed. The contribution of ion released is size dependent, it would be nice to estimate this dose. This is the missing link which would explains the observations.

 Answer: This is a very good point to be clarified in our manuscript. Exactly, the dissolution and ions release from the nanoparticles might have an impact on the induced cytotoxic effect specially for the smaller silver nanoparticles. We have added more explanation about this point in the discussion part. (Highlighted in yellow color lines 232-238).

We trust that these changes have addressed all of the reviewers’ comments. We hope that our revised manuscript is now suitable for publication.

Reviewer 2 Report

The MS is interesting; however, the following need to be addressed:

1- The MS needs to be reviewed by native english speaker. Some sentences need to be rephrased to be clearer.

2- It may be a good idea to support the findings by measuring cytokine production in macrophage cell line treated with TNF-a vs TNF-a + AgNPs.

3 - It may be a good idea to test the effect of AgNPs on a mouse model of inflammatory disease/s.

Author Response

Thank you for your comments. Please find the revised manuscript on the web site. Revisions have been made after carefully considering the comments raised by the reviewer.

Reviewer 2

1- The MS needs to be reviewed by native english speaker. Some sentences need to be rephrased to be clearer.

Answer: This MS has been taken grammar check by native speaker (see attached file).

 2- It may be a good idea to support the findings by measuring cytokine production in macrophage cell line treated with TNF-a vs TNF-a + AgNPs.

Answer: In this study, we only focused on the cellular responses of the lung epithelial cell line. However, testing in macrophage cell line would be highly recommended. We have added this point in our manuscript in the discussion part (Highlighted in yellow color lines 280-283).

 3 - It may be a good idea to test the effect of AgNPs on a mouse model of inflammatory disease/s.

Answer: This is a very helpful idea to support the in vitro studies with the findings of the in vivo studies using a laboratory animal model. We have clarified this point as a suggestion in our discussion. (Highlighted in yellow color lines 288-290).

 We trust that these changes have addressed all of the reviewers’ comments. We hope that our revised manuscript is now suitable for publication.

Round  2

Reviewer 2 Report

None